# MHC class I H2-K$^b$ negatively regulates neural progenitor cell proliferation by inhibiting FGFR signaling

Karin Lin[1,2]⊚, Gregor Bieri[1]⊚, Geraldine Gontier[1], Sören Müller[3], Lucas K. Smith[1,4], Cedric E. Snethlage[1], Charles W. White, III[1,5], Sun Y. Maybury-Lewis[6], Saul A. Villeda[1,2,4,5,7,8]*

1 Department of Anatomy, University of California San Francisco, San Francisco, California, United States of America, 2 Neuroscience Graduate Program, University of California San Francisco, San Francisco, California, United States of America, 3 Department of Neurological Surgery, University of California San Francisco, San Francisco, California, United States of America, 4 Biomedical Sciences Graduate Program, University of California San Francisco, San Francisco, California, United States of America, 5 Developmental and Stem Cell Biology Graduate Program, University of California San Francisco, San Francisco, California, United States of America, 6 Department of Molecular Biology, Cell Biology, and Biochemistry, Brown University, Providence, Rhode Island, United States of America, 7 Department of Physical Therapy and Rehabilitation Science, University of California San Francisco, San Francisco, California, United States of America, 8 The Eli and Edythe Broad Center for Regeneration Medicine and Stem Cell Research, San Francisco, California, United States of America

⊚ These authors contributed equally to this work.
* saul.villeda@ucsf.edu

**Data Availability Statement:** All quantitative data can be found in the supporting material (S1 Data).

## Abstract

Proteins of the major histocompatibility complex class I (MHC I), predominantly known for antigen presentation in the immune system, have recently been shown to be necessary for developmental neural refinement and adult synaptic plasticity. However, their roles in non-neuronal cell populations in the brain remain largely unexplored. Here, we identify classical MHC I molecule H2-K$^b$ as a negative regulator of proliferation in neural stem and progenitor cells (NSPCs). Using genetic knockout mouse models and in vivo viral-mediated RNA interference (RNAi) and overexpression, we delineate a role for H2-K$^b$ in negatively regulating NSPC proliferation and adult hippocampal neurogenesis. Transcriptomic analysis of H2-K$^b$ knockout NSPCs, in combination with in vitro RNAi, overexpression, and pharmacological approaches, further revealed that H2-K$^b$ inhibits cell proliferation by dampening signaling pathways downstream of fibroblast growth factor receptor 1 (Fgfr1). These findings identify H2-K$^b$ as a critical regulator of cell proliferation through the modulation of growth factor signaling.

## Introduction

Major histocompatibility class I (MHC I) proteins are most well known for their role in antigen presentation and immunological surveillance in the adaptive immune system [1]. More recently, in the central nervous system (CNS), immune components—such as complement,

Raw NSC RNA-sequencing data has been uploaded to ArrayExpress: E-MTAB-582.

**Funding:** This work was funded by California Institute for Regenerative Medicine predoctoral fellowship (K.L), Hillblom Foundation predoctoral fellowship (K.L), ARCS Foundation (L.K.S), Glenn Foundation (S.A.V) and the NIA (R01 AG067740, R01 AG055797). The funders had no role in study design, data collection and analysis, decision to publish, or preparation of the manuscript.

**Competing interests:** The authors have declared that no competing financial interests exist. LS is an associate editor for PLOS Biology. He was blinded to this manuscript and all related information in the journal's submission system and not involved at all in editorial discussions or the peer review process.

**Abbreviations:** BDNF, brain-derived neurotrophic factor; BrdU, 5-bromo-2′-deoxyuridine; CD68, cluster of differentiation 68; CNS, central nervous system; Dcx, Doublecortin; DG, dentate gyrus; EdU, 5-ethynyl-2′-deoxyuridine; Fgfr1, fibroblast growth factor receptor 1; GF, growth fraction; GFAP, glial fibrillary acidic protein; GFP, green fluorescent protein; GO, gene ontology; GWAS, genome-wide association studies; Iba1, ionized calcium-binding adapter molecule 1; IGF1, insulin-like growth factor 1; MHC I, major histocompatibility complex class I; NeuN, neuronal nuclei; NSC, neural stem cell; NSPC, neural stem and progenitor cell; PCC, Pearson correlation coefficient; RNAi, RNA interference; RNA-seq, RNA-sequencing; shRNA, short hairpin RNA; Tap1, transporter associated with antigen processing 1; Tbr2, T-box brain protein 2; VEGF, vascular endothelial growth factor; WT, wild-type.

cytokines, and MHC I molecules—have been demonstrated to play noncanonical roles in the developing and adult brain [2–5]. Specifically, classical MHC I molecules, H2-K$^b$ and H2-D$^b$, have been shown to negatively regulate visual system plasticity, neurite outgrowth, synapse density, and synaptic strength [6–11]. Notwithstanding, their role in regulating cellular functions beyond mature neuronal cell types in the brain remain largely unexplored.

Neural stem cells (NSCs), capable of producing functional neurons during embryonic and perinatal development as well as partially throughout life, represent a unique cell type critical in the formation of the CNS and potentially amenable to regenerative biology [12,13]. NSCs of the dentate gyrus (DG) reside in a specialized niche that supports their maintenance, proliferation, and differentiation into excitatory granule cells that integrate into hippocampal circuitry and confer plasticity [14]. This fine-tuned process is responsive to modulation by cell-intrinsic mechanisms, including transcription factors such as Sox2, TLX, and FoxOs, epigenetic factors like Tet2, posttranslational modifiers like Ogt, as well as changes to the neurogenic niche and systemic environment through manipulations such as exercise and exposure to young blood [15–21]. Importantly, neurogenesis can be regulated by immune system components such as cytokines, Toll-like receptors, and T cells [22–25]. One such factor, β2-microglobulin, the noncovalently associated subunit of the MHC I complex, has been implicated as a pro-aging systemic factor inhibiting neurogenesis [26]. However, the role of MHC I molecules themselves in regulating neural stem and progenitor cell (NSPC) function has yet to be investigated. Given the roles of MHC I molecules in negatively regulating neuronal processes and the susceptibility of NSPCs to immune influence, we sought to investigate the role of classical MHC I molecules, H2-K$^b$ and H2-D$^b$, in NSPC function.

In this study, we identify classical MHC I molecule H2-K$^b$ individually as a negative regulator of NSPC proliferation and hippocampal neurogenesis. Using a combination of in vivo genetic mouse models, viral-mediated in vivo RNA interference (RNAi) and overexpression, and in vitro NSPC functional assays, we identify H2-K$^b$ as a negative regulator of NSPC proliferation and adult hippocampal neurogenesis. Using RNA-sequencing (RNA-seq) transcriptomic analysis and in vitro RNAi, overexpression, and pharmacological inhibition, we provide evidence that loss of H2-K$^b$ in NSPCs enhances proliferation through fibroblast growth factor receptor 1 (Fgfr1) signaling pathways. Together, these data delineate a noncanonical role for a specific individual MHC I molecule, H2-K$^b$, in NSPC function.

## Results

### Loss of H2-K$^b$, and not H2-D$^b$, increases neurogenesis in the adult hippocampus

Previous studies have ascribed H2-K$^b$ and H2-D$^b$ with predominantly analogous functions in the brain [3]. Given the diversity of this family of molecules, we explored potential differential roles each individual classical MHC I molecule may play in NSPC function. To test this possibility, we first analyzed a publicly available dataset describing the transcriptomic dynamics of early phase adult hippocampal neurogenesis at the single-cell level [27]. Single-cell gene expression profiles were generated for Nestin-positive cells and their immediate progeny, with each cell bioinformatically assigned along a continuous trajectory from quiescent NSC activation to initiation of neurogenesis. Specifically, we divided the neurogenic lineage into an early time point (T1) where NSCs express glial fibrillary acidic protein (GFAP), and a later time point (T2) where differentiating intermediate neural progenitors begin to express T-box brain protein 2 (Tbr2) (**Figs 1A and S1**). Trend lines of *H2-K1*, *H2-D1*, and the proliferation marker *Ki67* expression were fitted along this neurogenic trajectory (**Fig 1A**). Interestingly, as *Ki67* expression increased *H2-K1* expression decreased, while *H2-D1* expression remained static

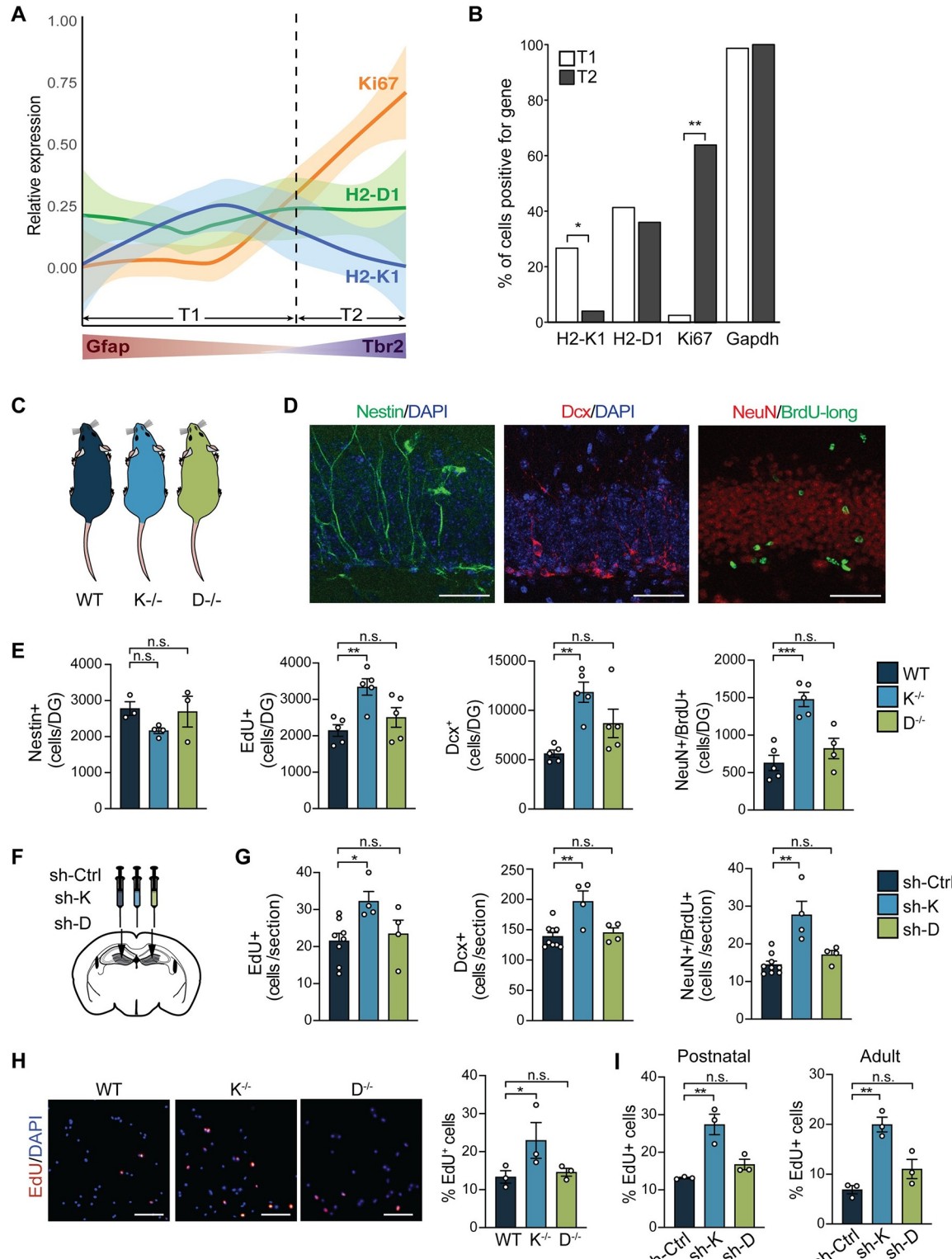

**Fig 1. H2-K^b, and not H2-D^b, negatively regulates adult hippocampal neurogenesis and NSPC proliferation.** Relative expression of *H2-K1* (H2-K^b), *H2-D1* (H2-D^b), and *Mki67* (Ki67) along a bioinformatically assigned trajectory of adult neurogenesis. Single adult hippocampal NSCs were ordered according to pseudotime (x-axis) estimates as given in [27]. Relative gene expression (y-axis, log2 (CPM+1)/max(log2(CPM+1))) was fitted with local polynomial regression fitting (bold lines) with 95% confidence interval (light colored area). The cutoff point between time points T1 and T2 was determined by the time of increasing Tbr2 expression (last

stationary point of the *Tbr2* regression line; see S1 Fig) (A). Number of cells positive for *H2-K1*, *H2-D1*, *Ki67*, and *Gapdh* expression (y-axis) between T1 and T2 (B). Neurogenesis was characterized in 3-month-old WT, H2-K$^b$ knockout (K$^{-/-}$), and H2-D$^b$ knockout (D$^{-/-}$) mice (C). Representative field (D) and quantifications (E) of Nestin$^+$ NSCs, EdU$^+$ short-term proliferating cells, Dcx$^+$ neuroblasts, and adult-born neurons coexpressing BrdU-long and NeuN in the DG. *n* = 3–5 mice/group (3–6 hippocampal sections/mouse); scale bars: 50 μm (D). Adult (3 months) WT mice were stereotaxically injected with lentivirus encoding an H1 promoter-driven shRNA targeting H2-K$^b$ (sh-K) or H2-D$^b$ (sh-D) in one DG and shRNA targeting luciferase as a control (sh-Ctrl) in the contralateral DG (F). (G) Quantification of EdU$^+$ proliferating cells (left panel), Dcx$^+$ neuroblasts (middle panel), and BrdU-long/NeuN double-positive mature neurons (right panel) in sh-K- and sh-D-injected DG compared to sh-control-injected DG (F). *n* = 4–8 DG/group (3–4 sections/DG). (H) Primary hippocampal WT, K$^{-/-}$, and D$^{-/-}$ NSPCs cultured under self-renewal conditions were treated with EdU for 6 hours. Representative field shown and mean percentage of EdU$^+$ dividing cells were quantified. *n* = 3 replicates/group; scale bars: 50 μm. (I) WT postnatal (left panel) and adult (right panel) NSPCs infected with lentiviruses encoding sh-K, sh-D, and sh-control were cultured under self-renewal conditions and treated with EdU for 6 hours. Percentage of EdU$^+$ postnatal and adult cells are shown. *n* = 3 replicates/group. Data are represented as mean ± SEM; Fisher's exact test (B); ANOVA with Dunnett's post hoc test (E, G, H, I); *$p < 0.05$, **$p < 0.01$, ***$p < 0.001$. Data used to generate this figure can be found in the Supporting information Excel spreadsheet (S1 Data). BrdU, 5-bromo-2′-deoxyuridine; Dcx, Doublecortin; DG, dentate gyrus; EdU, 5-ethynyl-2′-deoxyuridine; Gapdh, glyceraldehyde 3-phosphate dehydrogenase; Gfap, glial fibrillary acidic protein; NeuN, neuronal nuclei; n.s., not significant; NSC, neural stem cell; NSPC, neural stem and progenitor cell; shRNA, short hairpin RNA; Tbr2, T-box brain protein 2; WT, wild-type.

(**Fig 1A and 1B**). Correspondingly, the frequency of cells expressing *H2-K1* was significantly lower during T2 compared to T1, while no changes were observed for *H2-D1*, consistent with the housekeeping gene *Gapdh* (**Fig 1B**). These bioinformatics data suggest a potential role for H2-K$^b$ in regulating NSPC function during proliferative phase of adult neurogenesis.

To investigate the potential exclusivity of H2-K$^b$, and not H2-D$^b$, in regulating hippocampal neurogenesis in vivo, we utilized genetic mouse models in which *H2-K1* (K$^{-/-}$) or *H2-D1* (D$^{-/-}$) are individually excised (**Fig 2C**). Absence of *H2-K1* or *H2-D1* mRNA expression was confirmed by qPCR (**S2A Fig**). Cell proliferation was assessed by short-term 5-ethynyl-2′-deoxyuridine (EdU) labeling, and neurogenesis was examined by immunohistochemistry in adult K$^{-/-}$, D$^{-/-}$, and wild-type (WT) mice. The number of Nestin-positive radial glial-like stem cells was not different between genotypes (**Fig 1D and 1E**). However, absence of H2-K$^b$, but not H2-D$^b$, resulted in significantly more EdU-positive proliferating cells and Doublecortin (Dcx)-positive neuroblasts in the DG compared to WT controls (**Fig 1D and 1E**). Additionally, we assessed neuronal differentiation and survival using a long-term 5-bromo-2′-deoxyuridine (BrdU) incorporation paradigm, in which postmigratory differentiated neurons coexpress both BrdU and the mature neuronal marker neuronal nuclei (NeuN). The number of BrdU/NeuN double-positive differentiated mature neurons was increased in K$^{-/-}$, but not D$^{-/-}$, mice compared to WT controls (**Fig 1D and 1E**). Given H2-K$^b$ and H2-D$^b$ molecules are critical for immune cell development and function, we assessed microglia in the DG and detected no differences in the number of microglia expressing ionized calcium-binding adapter molecule 1 (Iba1) or the percent of microglia coexpressing the activation marker cluster of differentiation 68 (CD68) in K$^{-/-}$ or D$^{-/-}$ mice compared to WT controls (**S3A Fig**). No differences in GFAP-positive astrocytes or overall DG size were observed between genotypes (**S3B and S3C Fig**). Similar results for adult neurogenesis and microglia activation were observed in a genetic knockout mouse model in which *H2-K1* and *H2-D1* are concomitantly excised (KD$^{-/-}$; **S2B** and **S4A and S4B Figs**).

To complement the genetic mutant model and circumvent potential developmental effects of a constitutive knockout, we next used an in vivo viral-mediated RNAi approach. We generated lentiviral constructs encoding short hairpin RNA (shRNA) sequences selectively targeting either H2-K$^b$ (sh-K) or H2-D$^b$ (sh-D) mRNA, as well as targeting luciferase as a control (sh-Ctrl). Knockdown of H2-K$^b$ and H2-D$^b$ expression was confirmed by qPCR (**S2C Fig**). Adult WT mice were stereotaxically injected into the right DG with either sh-K or sh-D and into the left contralateral DG with sh-Ctrl (**Fig 1F**). Cell proliferation was assessed by short-term EdU labeling, and neuronal differentiation was assessed by long-term BrdU labeling. Local and

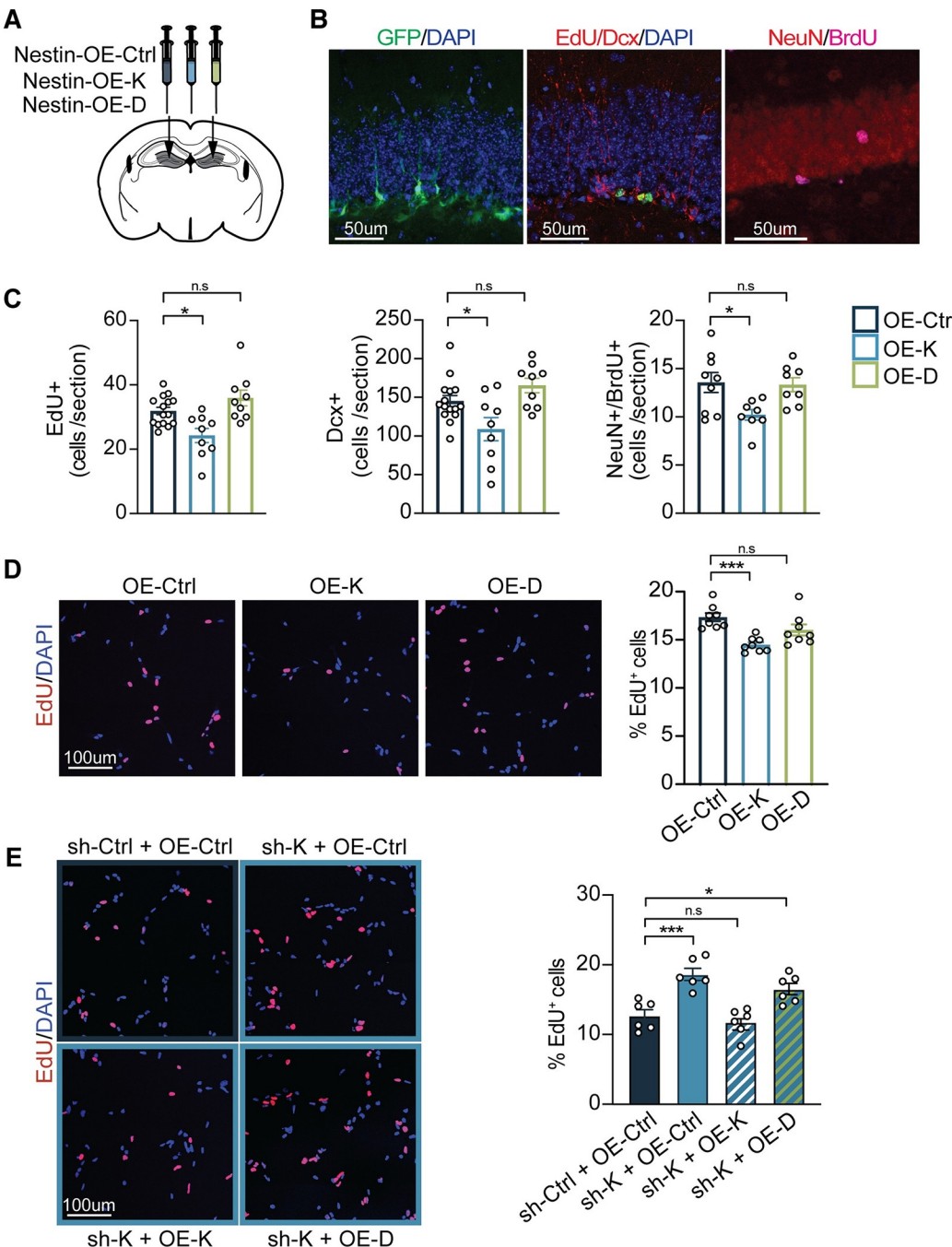

**Fig 2. H2-K$^b$ overexpression decreases NSPC proliferation and impairs adult hippocampal neurogenesis.** Ten-week-old adult WT mice were stereotaxically injected with lentiviruses expressing H2-K$^b$ (OE-K), H2-D$^b$ (OE-D), or GFP control (OE-Ctrl) under the Nestin promoter (A). (B) Representative fields of Nestin-GFP expressing cells (left panel), EdU$^+$ short-term proliferating cells, Dcx$^+$ neuroblasts (middle panel), and adult-born neurons coexpressing BrdU and NeuN (right panel) in the DG. (C) Quantification of EdU$^+$, Dcx$^+$, and BrdU$^+$/NeuN$^+$ cells in the DG. $n$ = 8–15 animals/group (3–4 sections/animal); scale bars: 50 μm. (D) WT adult NSPC were infected with OE-K, OE-D, and OE-Ctrl lentiviruses and grown under self-renewal conditions. Representative images (left panels) and quantification (right panel) of the percentage of EdU$^+$ proliferating NSPCs after 6 hours of EdU treatment. $n$ = 8 replicates/group; scale bars: 100 μm. (E) WT adult NSPC were infected with sh-K or sh-Ctrl, followed by infection with OE-K, OE-D, or OE-Ctrl lentiviruses. Representative images (E, left panel) and quantification (E, right panel) of EdU$^+$ proliferating NSPCs after 6 hours of EdU treatment. $n$ = 6 replicates/group; scale bar: 100 μm. Data represented as mean ± SEM. One-way ANOVA with Dunnett's multiple comparisons test; $^*p < 0.05$, $^{***}p < 0.001$. Data used to generate this figure can be found in the Supporting information Excel spreadsheet (S1 Data). BrdU, 5-bromo-2′-deoxyuridine; Dcx, Doublecortin; DG, dentate gyrus; EdU,

5-ethynyl-2′-deoxyuridine; GFP, green fluorescent protein; NeuN, neuronal nuclei; n.s., not significant; NSPC, neural stem and progenitor cell; WT, wild-type.

temporally controlled abrogation of H2-K$^b$ in the adult DG resulted in an increase in the number of EdU-positive proliferating cells, Dcx-positive neuroblasts, and BrdU/NeuN double-positive mature differentiated neurons compared to the contralateral control (**Fig 1G**). No differences were detected following abrogation of H2-D$^b$ in the adult DG compared to control conditions (**Fig 1G**). Together, these data identify H2-K$^b$ individually as a negative regulator of cell proliferation and neurogenesis in the adult hippocampus.

## Absence of H2-K$^b$ in NSPCs increases proliferation, in part, by altering cell cycle dynamics

Having observed the involvement of H2-K$^b$ in regulating adult neurogenesis in vivo, and to circumvent the lack of cell-type specificity inherent in using constitutive knockout or viral-mediated RNAi in vivo models, we next investigated the functional role of H2-K$^b$ in NSPC proliferation in vitro. We subdissected hippocampi from K$^{-/-}$, D$^{-/-}$, and WT mice and cultured primary postnatal NSPCs following an established protocol [28]. K$^{-/-}$, D$^{-/-}$, and WT NSPCs were cultured under self-renewal conditions and treated with EdU. We detected a significant increase in the percentage of EdU-positive proliferating cells in K$^{-/-}$ compared to WT NSPCs (**Fig 1H**), while no differences were observed in D$^{-/-}$ NSPCs (**Fig 1H**). Under differentiation conditions in vitro, no differences in the percentage of neurons and astrocytes were detected between genotypes (**S5A and S5B Fig**), suggesting a role for H2-K$^b$ in specifically regulating the proliferative process in NSPCs in vitro. These results were corroborated by in vitro studies in cultured WT postnatal and adult NSPCs, where acute abrogation of H2-K$^b$, but not H2-D$^b$, expression by RNAi resulted in a significant increase in EdU-positive proliferating cells compared to control conditions (**Fig 1I**). No differences in differentiation were observed following H2-K$^b$ or H2-D$^b$ abrogation (**S5C Fig**). Increased NSPC proliferation was also observed in WT postnatal and adult NSPCs in which H2-K$^b$ and H2-D$^b$ expression is concomitantly decreased (**S2D** and **S4C and S4D Figs**). These in vitro data identify H2-K$^b$ as a negative regulator of NSPC proliferation.

To further examine the role of H2-K$^b$ in regulating adult NSCP proliferation, we made use of the publicly available dataset described above (**Fig 1A and 1B**). We scored each single cell analyzed for expression of signatures indicative of the G1/S or G2/M phases of the cell cycle (as described in Tirosh and colleagues [29]) and subsequently identified each cell as either positive or negative for *H2-K1* expression (**S6A Fig**). We found that the percentage of cycling cells to be lower in *H2-K1*-positive cells (5.6%) versus *H2-K1*-negative cells (24.4%), suggesting that adult NSCs and their immediate progeny are more likely to be noncycling when enriched for *H2-K1* (**S6B Fig**). These bioinformatics data provide evidence that H2-K$^b$ potentially regulates NSPC proliferation by impacting the cell cycle.

Increased hippocampal NSPC proliferation is a phenomenon known to occur under certain stimuli; in the case of seizures, a shortened cell cycle length is the cellular mechanism underlying the enhancements in cell proliferation [30]. Correspondingly, we examined if similar changes occurred in cell cycle dynamics of NSPCs lacking H2-K$^b$. Primary hippocampal NSPCs isolated from K$^{-/-}$ and WT mice were cultured under self-renewal conditions and were subsequently treated with EdU at 2-hour intervals over a 20-hour period. Absence of H2-K$^b$ resulted in an increase in the ratio of proliferating cells to the total number of cells in the population, corresponding to the growth fraction (GF; **S6C Fig**). Furthermore, we detected

a decrease in both the length of the S-phase ($T_s$) and length of the cell cycle ($T_c$; **S6C Fig**), calculated as described in Ponti and colleagues [31]. These in vitro data indicate that NSPCs lacking H2-$K^b$ exhibit increased proliferation, in part, due to an acceleration of cell cycle progression.

## Overexpression of H2-$K^b$, and not H2-$D^b$, in neural stem cells decreases adult hippocampal neurogenesis and impairs NSPC proliferation

To complement our loss of function studies, we investigated the effect of increasing H2-$K^b$ expression in adult NSCs. We used a cell-type specific in vivo viral-mediated overexpression approach, in which we generated lentiviral constructs encoding either H2-$K^b$ (OE-K), H2-$D^b$ (OE-D), or green fluorescent protein (GFP) as a control (OE-ctrl) under the control of the Nestin promoter. Overexpression of *H2-K1* and *H2-D1* expression was confirmed by qPCR (**S2E Fig**). Adult WT mice were stereotaxically injected into the right DG with either OE-K or OE-D and into the left contralateral DG with OE-ctrl (**Fig 2A**). Cell proliferation was assessed by short-term EdU labeling, and neuronal differentiation was assessed by long-term BrdU labeling. Increased expression of H2-$K^b$ in adult NSCs resulted in a decrease in the number of EdU-positive proliferating cells, Dcx-positive neuroblasts, and BrdU/NeuN double-positive mature differentiated neurons compared to the contralateral control (**Fig 2B and 2C**). No differences were detected following overexpression of H2-$D^b$ in adult NSCs compared to control conditions (**Fig 2B and 2C**).

We next investigated the effect of increased H2-$K^b$ expression on adult NSPC proliferation in vitro. WT adult NSPCs were infected with OE-K, OE-D, or OE-ctrl lentivirus and treated with EdU. Overexpression of H2-$K^b$, but not H2-$D^b$, resulted in a significant decrease in EdU-positive proliferating cells compared to control (**Fig 2D**). To further delineate the role of H2-$K^b$, we examined whether the effects of abrogating H2-$K^b$ on NSPC proliferation could be rescued by overexpressing either H2-$K^b$ or H2-$D^b$. WT adult NSPC were infected with sh-K lentivirus in combination with OE-K, OE-D, or OE-ctrl. As a negative control, WT adult NSPC were infected with sh-Ctrl and OE-ctrl lentivirus. Consistent with our previous findings (**Fig 1I**), abrogation of H2-$K^b$ resulted in an increase in EdU-positive proliferating cells (**Fig 2E**); however, this effect was mitigated by overexpression of H2-$K^b$ but not H2-$D^b$ (**Fig 2E**). Collectively, these in vivo and in vitro data indicate that H2-$K^b$ is sufficient to impair adult hippocampal neurogenesis and NSPC proliferation.

## H2-$K^b$ regulates NSPC proliferation by inhibiting Fgfr-mediated signaling

To gain mechanistic insight into how MHC I molecules regulate NSPC proliferation, we profiled gene expression in primary NSPCs derived from $K^{-/-}$, $D^{-/-}$, and WT mice using RNA-seq. Hierarchical clustering revealed high similarity between biological replicates within individual genotypes (average Pearson correlation coefficient; PCC >0.99). Interestingly, replicates from $K^{-/-}$ NSPCs were segregated from all other genotypes (**Fig 3A**). As expected, we found a larger number of differentially expressed genes (844 genes) between $K^{-/-}$ and WT NSPCs compared to $D^{-/-}$ and WT (118 genes) NSPCs (**S7A Fig**). Next, we focused on genes whose expression was significantly different between WT and $K^{-/-}$ NSPCs (**S7B Fig**). Among the most differentially up-regulated genes detected in $K^{-/-}$ NSPCs, we identified several growth factor and extracellular matrix protein encoding genes (**Fig 3B**); many of which are associated with increased NSPC proliferation [12]. Accordingly, we found "cell proliferation" among the most significantly enriched gene ontology (GO) terms (**Fig 3C**).

To elucidate pathways that might underlie increased proliferation observed in $K^{-/-}$ NSPCs, we made use of our RNA-seq dataset and performed pathway enrichment analysis (**Fig 3C**).

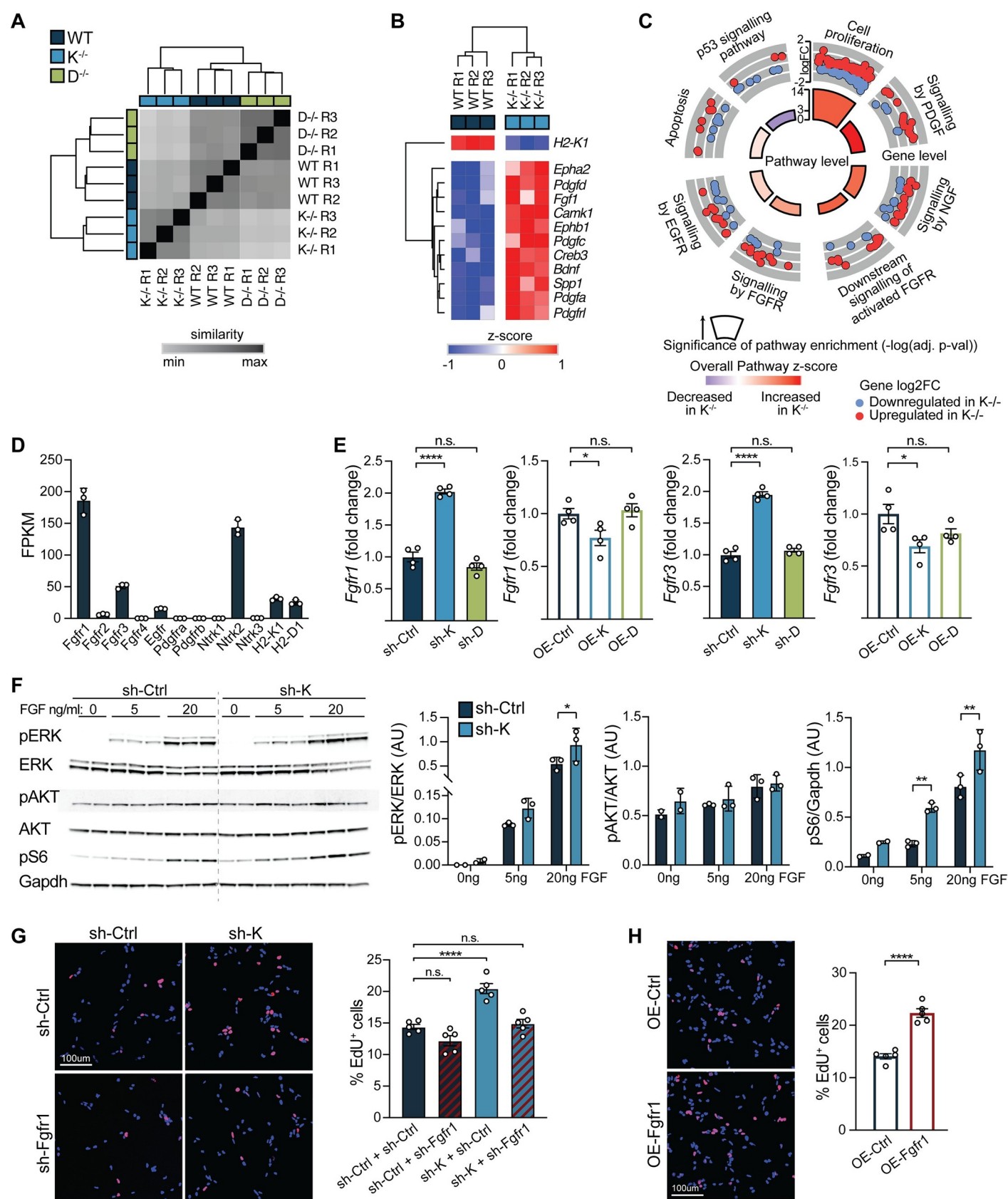

**Fig 3. H2-K$^b$ negatively regulates NSPC proliferation by inhibiting Fgfr signaling.** RNA-seq analysis of primary hippocampal WT, K$^{-/-}$, and D$^{-/-}$ NSPCs cultured under self-renewal conditions. Pairwise distances (Euclidean distance) between RNA-seq libraries prepared from WT, K$^{-/-}$, and D$^{-/-}$ primary hippocampal NSPCs. Dendrogram was calculated by unsupervised hierarchical clustering (average linkage). ($n$ = 3 replicates/group [denoted R1–R3]) (A). (B) Unsupervised hierarchical clustering of growth factor signaling-related genes identified to be differentially expressed between WT and K$^{-/-}$ hippocampal NSPCs using RNA-seq analysis (z-score normalized, rows). (C) Cellular and signaling pathways identified by pathway enrichment analysis (ConsensusPathDB) of genes differentially expressed between WT and K$^{-/-}$ NSPCs are subdivided in gray. Outer ring illustrates expression of altered genes (log2 fold change between K$^{-/-}$ vs WT) within each pathway. Inner ring illustrates significance (−log10 adj. $p$-value) and overall expression levels of genes within each pathway (z-score). (D) Expression (FPKM) of growth factor receptors and H2-K1 and H2-D1 was evaluated in WT NSPCs using RNA-seq. $n$ = 3 replicates/group for all RNA-seq analysis. (E) WT adult NSPCs were infected with either sh-K and sh-control (sh-Ctrl) or Nestin-driven K overexpression (OE-K) and GFP control (OE-Ctrl) lentiviruses. $Fgfr1/Fgfr3$ expression were measured using quantitative RT-PCR 72 hours after infection. $n$ = 4 replicates/group. (F) Representative western blot (left panel) and quantifications (right panels) of activated Akt, Erk, and ribosomal protein S6 in adult WT NSPCs infected with lentiviruses encoding sh-K or sh-control (sh-Ctrl). NSPCs were grown in the absence of (0 ng/mL), low (5 ng/mL), or regular (20 ng/mL) FGF level for 2 hours. $n$ = 2–3 replicates/time point/group; data represented as mean ± SD. (G) Representative images (left panel) and quantification (right panel) of adult NSPCs infected with sh-K or sh-Ctrl in combination with lentiviruses expressing shRNAs against $Fgfr1$ (sh-Fgfr1) or control (sh-Ctrl). NSPCs were grown under self-renewal conditions and treated with EdU for 6 hours to test for proliferation. Percentage of EdU$^+$ cells are shown. $n$ = 5 replicates/group. (H) Representative images (left panel) and quantification (right panel) of WT adult NSPCs infected with Nestin-Fgfr1 overexpression (OE-Fgfr1) or GFP control (OE-Ctrl) lentiviruses. NSPCs and cultured under self-renewal conditions treated with EdU for 6 hours to test for proliferation. Percentage of EdU$^+$ cells are shown. $n$ = 5 replicates/group. Data represented as mean ± SEM. One-way ANOVA with Dunnett's multiple comparisons test (E, G), two-way ANOVA with Sidak's multiple comparisons test (F), and Student $t$ test (H). $^*p < 0.05$, $^{**}p < 0.01$, $^{****}p < 0.0001$. Data used to generate this figure can be found in the Supporting information Excel spreadsheet (S1 Data). EdU, 5-ethynyl-2′-deoxyuridine; EGFR, epidermal growth factor receptor; FGF, fibroblast growth factor; Fgfr, fibroblast growth factor receptor; Fgfr1, fibroblast growth factor receptor 1; Gapdh, glyceraldehyde 3-phosphate dehydrogenase; GFP, green fluorescent protein; NGF, nerve growth factor; n.s., not significant; NSPC, neural stem and progenitor cell; PDGF, platelet-derived growth factor; RNA-seq, RNA-sequencing; RT-PCR, reverse transcription polymerase chain reaction; shRNA, short hairpin RNA; WT, wild-type.

Genes up-regulated in K$^{-/-}$ NSPCs comprised several growth factor receptor tyrosine kinase signaling pathways (Fig 3C). Apoptosis pathway-associated genes did not differ between groups, whereas p53 tumor suppressor pathway genes were down-regulated in K$^{-/-}$ NSPCs (Fig 3C). As multiple categories of growth factor signaling were enriched, we next surveyed growth factor receptor expression in WT NSPCs in our RNA-seq dataset and detected highest expression of $Fgfr1$ followed by $Ntrk2$, $Fgfr3$, $Egfr$, and $Fgfr2$ (Fig 3D). Next, we surveyed growth factor receptor expression in WT adult NSPCs following viral-mediated abrogation and overexpression of H2-K$^b$ or H2-D$^b$ by qPCR (Figs 3E and S8). We observed a selective bidirectional change in $Fgfr1$ and $Fgfr3$ in WT NSPCs infected with sh-K and OE-K lentivirus, in which $Fgfr1$ and $Fgfr3$ expression increased following H2-K$^b$ abrogation and decreased following H2-K$^b$ overexpression (Fig 3E). Correspondingly, we examined whether downstream growth factor receptor tyrosine kinase signaling pathways were more highly activated following H2-K$^b$ abrogation. We assessed Akt and Erk pathways in WT adult NSPCs infected with sh-K and grown under absent (0 ng/mL), low (5 ng/mL), or normal (20 ng/mL) FGF conditions. Western blot analysis showed that decreased H2-K$^b$ expression enhanced phosphorylation of Erk (Thr202/Tyr204) but not Akt (Tyr450) (Fig 3F). Ribosomal protein S6, a downstream target of receptor tyrosine kinase signaling involved in cell proliferation, was also more highly phosphorylated (Ser235/236) following H2-K$^b$ abrogation (Fig 3F). These data suggest that H2-K$^b$ negatively regulates NSPC proliferation through inhibition of receptor tyrosine kinase signaling pathways downstream of Fgfr signaling.

Given that Fgfr1 showed highest growth factor receptor expression in WT NSPCs (Fig 3D) and exhibited bidirectional gene expression changes following H2-K$^b$ abrogation and overexpression (Fig 3E), we opted to investigate the functional role of Fgfr1 in mediating the effects of decreased H2-K$^b$ expression on cell proliferation. We generated lentiviral constructs encoding shRNA sequences targeting Fgfr1 (sh-Fgfr1) and knockdown was confirmed by qPCR (S9A Fig). WT adult NSPC were infected with sh-K or sh-Ctrl lentivirus in combination with sh-Fgfr1 or sh-Ctrl and treated with EdU. Consistent with our previous findings (Figs 1I and 2E), decreased H2-K$^b$ expression resulted in an increase in EdU-positive proliferating cells (Fig 2E); however, this effect was mitigated by abrogation of Fgfr1 (Fig 3G). Similar findings were observed in vitro following treatment with an FGFR pharmacological inhibitor (S10 Fig).

Last, we examined whether mimicking the increase in Fgfr1 expression observed following H2-K$^b$ abrogation (**Fig 3E**) could likewise increase in NSPC proliferation. We generated lentiviral constructs encoding Fgfr1 (OE-Fgfr1) under the control of the Nestin promoter and overexpression was confirmed by qPCR (**S9B Fig**). Increased expression of Fgfr1 resulted in a significant increase in EdU-positive proliferating cells compared to control (**Fig 2D**). Altogether, these data indicate that H2-K$^b$ molecules negatively regulate NSPC proliferation by inhibiting Fgfr1-mediated signaling.

## Discussion

In this study, we identify classical MHC I molecule H2-K$^b$ as a negative regulator of hippocampal neurogenesis. Using genetic knockout mouse models, viral-mediated RNAi and overexpression approaches, and primary NSPC cultures, we demonstrate that H2-K$^b$ negatively regulates adult neurogenesis and NSPC proliferation. Subsequent transcriptomic analysis of WT and K$^{-/-}$ NSPCs, in combination with in vitro RNAi, overexpression, and pharmacological approaches, revealed that H2-K$^b$ inhibits cell proliferation by dampening signaling pathways downstream of Fgfr1. Collectively, our findings attribute H2-K$^b$ a previously unrecognized inhibitory role in NSPC function. As many CNS disorders with neurogenic dysfunction also coexhibit changes in MHC I expression [32–34], our findings provide a unique molecular target to promote the innate regenerative capacity of the CNS to combat such brain insults.

Previous studies investigating classical MHC I molecules in the CNS have utilized broad genetic manipulations, where β2-microglobulin and transporter associated with antigen processing 1 (Tap1), in which MHC I molecules cannot be trafficked to the cell membrane, were constitutively deleted in the whole body. These models targeted MHC I as a whole class of molecules, as they decreased global cell surface expression of most, if not all, murine MHC I proteins indiscriminately. Due to the available genetic models, there has been a general assumption that classical MHC I subtypes performed analogous functions in the brain. However, as MHC I molecules comprise of a highly polymorphic gene family with over 50 sequences described, it is critical to begin to elucidate specific roles and cell-type significance for individual MHC I molecules in the CNS. Recently, it was shown that reintroduction of neuronal H2-D$^b$ selectively could reverse synaptic deficits in KD$^{-/-}$ mice, suggesting that a single MHC I molecule was sufficient to rescue functional synapse elimination and refinement in the visual system [35]. Concordantly, we examined involvement of H2-K$^b$ and H2-D$^b$ separately in single knockout models and using NSC-specific viral-mediated overexpression approaches. While we do not exclude a role for H2-D$^b$ in NSPC function, our data reveal a distinct role for H2-K$^b$ in directing NSPC proliferation and hippocampal neurogenesis. Additionally, in this study, spatially and temporally controlled H2-K$^b$ and H2-D$^b$ abrogation using RNAi in WT mice further allowed us to tease apart the general effects of systemic immune compromise in genetic knockout models from absence of H2-K$^b$ and H2-D$^b$ selectively in the hippocampus. The functional difference observed between H2-K$^b$ and H2-D$^b$ may stem from disparities in structure, spatiotemporal distribution, or binding partners, and future studies are required for detailed comparative characterization.

Our study also provides the first unbiased transcriptomic insight into pathways that may lie downstream of MHC I in the CNS. RNA-seq analysis revealed enrichment of several growth factor receptor signaling pathways in K$^{-/-}$ NSPCs, of which Fgfr1 exhibited bidirectional gene expression changes following H2-K$^b$ abrogation and overexpression. Mechanistically, we further demonstrate that H2-K$^b$ molecules negatively regulate NSPC proliferation by inhibiting Fgfr1-mediated signaling. To date, studies have shown that a number of soluble growth factors,

such as insulin-like growth factor 1 (IGF1), vascular endothelial growth factor (VEGF), brain-derived neurotrophic factor (BDNF), FGF2, and EGF regulate adult neurogenesis [12]. Interestingly, it has recently been demonstrated that restoring growth factor signaling by increasing FGF2 or EGF in the neurogenic niche or activating Fgfr in adult NSPCs is sufficient to enhance adult neurogenesis in rodents where aging or neurodegenerative disease has caused functional decline of stem cell function [36–38]. It is promising that despite pathological neurogenic decline, the remaining endogenous NSPCs are amenable to rejuvenation through activation of growth factor signaling. Further insights into precisely how interactions between H2-K$^b$ and growth factor receptors affect their trafficking dynamics, internalization, expression, and signaling specifically on NSPCs will be required to better understand how H2-K$^b$ can be therapeutically targeted under conditions of aging and neurodegenerative disease.

Under physiological conditions, neurogenesis has been implicated in spatial learning and memory, pattern separation, and affective behaviors [39]. Although tightly controlled, this process is highly vulnerable under pathological conditions: In mammalian models, NSPC loss of function occurs in aging and neurodegenerative disorders, while aberrant gain of NSPC proliferation can occur in epilepsy and schizophrenia [14,40]. Preventing NSPC dysfunction may be essential for restoring brain function under pathological conditions, and identification of specific molecules and mechanisms controlling neurogenesis will inform translational efforts. Our finding that H2-K$^b$ uniquely regulates NSPC proliferation suggests that classical MHC I molecules exhibit polymorphic biological functions in the brain. This posits that certain individual MHC I variants may selectively modulate susceptibility for neurological disorders. Indeed, MHC I molecules have been implicated in several CNS and systemic diseases—many of which present with NSPC dysfunction—by human genome-wide association studies (GWAS). Single nucleotide polymorphisms on the MHC complex region on chromosome 6 have been linked with highly significant disease risk in schizophrenia and age-associated disorders [41–44]. In addition to a genetic association to disease, MHC I expression has directly been shown to be dynamic in certain pathophysiological states: MHCI molecules increase in rodent hippocampal neurons and neuromuscular junctions with age, interfering with synaptic homeostasis [32,34], and an up-regulation of neuronal H2-K$^b$, H2-D$^b$, and MHC I receptor, PirB, is observed in stroke [33]. As animals lacking H2-K$^b$ and H2-D$^b$ are protected from tissue damage and exhibit enhanced motor recover after stroke [33], it remains for future studies to now test whether targeting H2-K$^b$ molecules in the aging hippocampus could prevent, or even reverse, age-related neurogenic decline.

## Methods

### Experimental model and subject details

All experiments were randomized and blinded by an independent researcher prior to pharmacological treatment or assessment of genetic mouse models. Researchers remained blinded throughout histological and biochemical assessments. Groups were unblinded at the end of each experiment upon statistical analysis. A key resource table can be found in the Supporting information (S1 Table).

### Mouse models

All animal handling and use was approved by the Institutional Animal Care and Use Committee at the University of California San Francisco (AN178706-03B), in compliance with all applicable provisions of the Animal Welfare Act and Regulations (AWAR), the Public Health Service (PHS) Policy on Humane Care and Use of Laboratory Animals, and other federal regulations and policies relating to the use of animals in research and instruction. All mice were

group housed under specific pathogen-free conditions under a 12-hour light–dark cycle with ad libitum access to food and water.

The following mouse lines were used: C57BL/6 (Taconic), *H2-K* and *H2-D1* (KD$^{-/-}$) double knockout mice (Taconic), *H2-K1* (K$^{-/-}$) single knockout mice (Taconic), and *H2-D1* (D$^{-/-}$) single knockout mice (Taconic), and C57BL/6J (The Jackson Laboratory). C57BL/6J mice were used for all stereotaxic injection experiments. All studies were conducted using male animals. The number of mice used to result in statistically significant differences was calculated using standard power calculations with α = 0.05 and a power of 0.8. We used an online tool (http://www.stat.uiowa.edu/~rlenth/Power/index.html) to calculate power and samples size based on experience with the respective tests, variability of the assays, and interindividual differences within groups.

## Immunohistochemistry

Tissue processing and immunohistochemistry were performed on free-floating sections following standard published techniques [26]. Briefly, mice were anesthetized with a ketamine (100 mg/kg)-xylazine (10 mg/kg) cocktail (Patterson Veterinary, Henry Schein) and transcardially perfused with cold PBS. Brains were removed, fixed in phosphate-buffered 4% paraformaldehyde (pH 7.4), at 4°C for 48 hours followed by cryoprotection in 30% sucrose, and coronally sectioned at 40 μm with a cryomicrotome (Leica SM2010 R). Sections were washed 3 times in Tris-buffered saline with 0.1% Tween 20 (TBST) and incubated in 3% normal donkey serum (Thermo Fisher Scientific) for 1 hour. After overnight incubation in primary antibody (see S1 Table) at 4°C, staining was revealed using fluorescence conjugated secondary Alexa antibodies (1:500; see S1 Table). Antigen retrieval for BrdU labeling required incubation in 3M HCl at 37°C for 30 minutes before incubation with primary antibody; Nestin labeling required incubation in citrate buffer (Sigma-Aldrich Cat# C9999-100ML) at 95°C (3 times 5 minutes) prior to incubation with primary antibody. For EdU labeling, brain sections were incubated with freshly made EdU reaction mixture for 45 minutes at room temperature following the manufacturer's instructions. Then, sections were washed, blocked, and incubated with primary antibodies following the regular immunostaining protocol. To estimate the total number of immunopositive cells per DG, confocal stacks of coronal sections of the DG (4 to 6 sections per mouse, 40 μm thick, 240 μm apart) were acquired on a Zeiss LSM 800. Immunopositive cells in the granule and subgranular cell layers were then counted and multiplied by 12 to estimate the total number in the entire DG.

## Western blot analysis

Primary NSCs were lysed in prechilled RIPA lysis buffer (500 mM Tris (pH 7.4), 150 mM NaCl, 0.5% Na deoxycholate, 1% NP40, 0.1% SDS) supplemented with complete protease inhibitor cocktail (Sigma cat# 11697498001) and Halt phosphatase inhibitor cocktail (Thermo Fisher Scientific cat# 78420). Cell lysates were incubated for 15 minutes on ice and cleared using 10-minute 15,000 rpm centrifugation at 4C. Protein concentrations were measured using a BCA protein assay kit (Thermo Fisher Scientific cat# PI23225). Protein lysates were mixed with 4× NuPage LDS loading buffer (Invitrogen), heat denatured, loaded on a 4% to 12% SDS polyacrylamide gradient gel (BioRad cat # 3450124), and subsequently transferred onto a nitrocellulose membrane. The blots were blocked in 5% milk in TBST and incubated with primary antibody at 4°C for 16 hours. Horseradish peroxidase–conjugated secondary antibodies and an ECL kit (BioRad cat # 1705060 and 1705062) were used to detect protein signals. Multiple exposures were taken on a BioRad ChemiDoc. Images were exported (300 dpi)

and quantified using the Gels tool in FIJI/ImageJ (Version 2.1.0). GAPDH, total Erk, and Akt bands were used for normalization.

## RNA isolation and qRT-PCR

mRNA of NSCs was isolated by lysis with TRIzol Reagent (Thermo Fisher Scientific), separation with chloroform (0.2 mL per mL TRIzol), and precipitated with isopropyl alcohol. mRNA from DG tissue was isolated with the Rneasy Mini Kit (Qiagen). High Capacity cDNA Reverse Transcription Kit (Life Technologies) was used for reverse transcription of mRNA into cDNA, and qRT-PCR was carried out using Power SYBR Green PCR Master Mix (Life Technologies) in a CFX384 Real Time System (Bio-Rad). Primers were: $H2\text{-}K^b$ forward 5′-CGGCGCTGAT CACCAAACA-3′, $H2\text{-}K^b$ reverse 5′-AGCGTCGCGTTCCCGTT-3′; $H2\text{-}D^b$ forward 5′ -GTA AAGCGTGAAGACAGCTGC- 3′, $H2\text{-}D^b$ reverse 5′-CTGAACCCAAGCTCACAGG-3′ (3′UTR); $H2\text{-}D^b$ forward-2 5′—3′; $H2\text{-}D^b$ reverse-2 5′—3′ (CDS); GAPDH forward 5′-GCAT CCTGCACCACCAACTG-3′, GAPDH reverse 5′-ACGCCACAGCTTTCCAGAGG-3′. Fgfr1 forward 5′-CTTGCCGTATGTCCAGATCC-3′; Fgfr1 reverse 5′-TCCGTAGATGAAGC ACCTCC-3′; Fgfr2 forward 5′-TGCATGGTTGACAGTTCTGC-3′; Fgfr2 reverse 5′-GCAG GCGATTAAGAAGACCC-3′; Fgfr3 forward 5′-GAAGCACGTGGAAGTGAACG-3′; Fgfr3 reverse 5′-TCCTTGTCGGTGGTGTTAGC-3′; Egfr forward 5′-GGACTGTGTCTCCTGCCA GAAT-3′; Egfr reverse 5′-GGCAGACATCCTGGATGGCACT-3′; Ntrk2 forward 5′-TTACG TGGGGCTGAGAAACC-3′; Ntrk2 reverse 5′-TCCTGGACAAACTCGTCAGC-3′. All primer pairs display a single peak using melting curve analysis.

## BrdU/EdU administration and quantification

For short-term proliferation studies, mice were intraperitoneally injected with Brdu (Brdu-short) or EdU (50 mg/kg body weight, Thermo Fisher Scientific) 5 days before euthanasia. For study of newborn neuron survival, mice were injected with BrdU (BrdU-long) for 5 consecutive days, and animals were euthanized 28 days after administration. Stereotaxically injected animals received their first injection after a minimum of 2 weeks to 1 month of recovery from surgery. EdU labeling was revealed using the Click-it kit (Thermo Fisher Scientific) following the manufacturer's instruction. To estimate the total number of BrdU-positive cells in the brain, we performed fluorescence staining for BrdU on 3 to 6 hemibrain sections per mouse (40 μm thick, 240 μm apart). The number of EdU/BrdU-positive cells in the granule cell and subgranular cell layer of the DG was counted and multiplied by 12 to estimate the total number of EdU/BrdU-positive cells in the entire DG. To quantify neuronal fate and maturation of dividing cells, BrdU-positive cells across 4 to 6 sections per mouse were analyzed by confocal microscopy for coexpression with NeuN. For stereotaxially injected animal, 3 to 4 brain sections without damage, surrounding the injection site were analyzed, and cell numbers were computed as the average number of cells/section.

## Stereotaxic injections

Animals were placed in a stereotaxic frame and anesthetized with 2% isoflurane (Patterson Veterinary) (2 L per min oxygen flow rate) delivered through an anesthesia nose cone. Fur around the incision area was trimmed, and ophthalmic eye ointment was applied to the cornea to prevent desiccation during surgery. Viral solutions were injected bilaterally into the dorsal hippocampi using the following coordinates: (from bregma) anterior = −2 mm, lateral = 1.5 mm and (from skull surface) height = −2.1 mm. A 2-μL volume of viral solution was injected stereotaxically over 10 minutes (injection speed: 0.20 μL per min) using a 5-μL 26s-gauge Hamilton syringe. To limit reflux along the injection track, the needle was maintained in situ

for 8 minutes, slowly pulled out halfway, and kept in position for an additional 2 minutes. The skin was closed using silk suture. Each mouse was injected subcutaneously with enrofloxacin (Bayer) antibiotic and buprenorphine (Butler Schein) as directed for pain, single housed, and monitored during recovery. After a minimum of 2 weeks of recovery, animals were injected with BrdU for 5 consecutive days and euthanized 28 days following the final injection. During the last 5 days prior to euthanasia, all animals were intraperitoneally injected with EdU.

## Isolation of primary neural stem cells from the postnatal and adult mouse hippocampus

Postnatal and adult primary NSC isolation and culture were preformed following previously published techniques [28]. Hippocampi were dissected from postnatal day 1 (P1) WT C57/ BL6 control, KD$^{-/-}$, K$^{-/-}$, or D$^{-/-}$ mice, or adult (2 months) WT C57/BL6 mice and pooled by genotype for NSPC isolation without distinguishing genders. After removing superficial blood vessels, hippocampi were mechanically dissociated by fine mincing and enzymatically digested for 30 minutes at 37°C in DMEM media containing 2.5 U/ml Papain (Worthington Biochemicals), 1 U/ml Dispase II (Boeringher Mannheim), and 250 U/ml Dnase I (Worthington Biochemicals). NSCs were purified using a 65% Percoll gradient and cultured (Neural Basal A medium supplemented with 2% B27 without Vitamin A, 1% Glutamax, 1% Penicillin Streptomycin, 10 ng/ml EGF, 10 ng/ml bFGF) as a monolayer on poly-D-lysine and laminin-coated plates at a density of $10^5$ cells/cm$^2$.

## NSC proliferation assay

WT or knockout NSPCs were seeded in 500 μL growth medium (Neural Basal A medium supplemented with 2% B27 without Vitamin A, 1% Glutamax, 1% Penicillin Streptomycin, 10 ng/ml EGF, 10 ng/ml bFGF) at a density of 10,000 cells/well on PDL/laminin-coated glass coverslips in a 24-well tissue culture plate. Twenty-four hours later, cells were treated with 20 μM BrdU or EdU for 6 hours prior to fixing with 4% PFA. Acute knockdown of H2-K$^b$, H2-D$^b$, or both was induced by treating plated WT NSCs with shRNA-encoding lentiviruses ($6.5 \times 10^6$ to $7.5 \times 10^6$ viral particles per mL media) for 3 days prior to BrdU or EdU treatment as described. For experiments requiring a combination of 2 different viruses, only half the concentration of virus was added to the media for each virus. Cells were washed 3 times with PBS after fixation, blocked in 3% donkey serum, and incubated in primary antibody at 4°C for 16 hours. After 3 washes with PBS, BrdU staining was revealed with fluorescence-conjugated secondary Alexa antibodies (1:500; see S1 Table). EdU incorporation was revealed using a Click-iT EdU Alexa Fluor Imaging Kit (Thermo Fisher Scientific), and nuclei were counterstained with Hoechst 33342 (1:10,000; Thermo Fisher Scientific). BrdU- or EdU-positive cells were counted per field of view at 3 to 4 randomly determined locations per coverslip. In vitro experiments were conducted in triplicates for each condition (unless stated otherwise) and repeated to ensure reproducibility. For inhibitor experiments, NSPCs were pretreated for 1 hour with an Fgfr inhibitor (PD 161570, 0.1uM) prior to the 6-hour EdU pulse.

## Cell cycle analysis by cumulative EdU labeling

WT and K$^{-/-}$ NSCs cultured in vitro in growth media were treated with 20 μM EdU at 2, 4, 6, 8, 10, 12, 14, 16, 18, 20, 22, or 24 hours prior to fixation. Cells that incorporated EdU were labeled using the Click-iT Plus EdU Alexa Fluor Flow Cytometry Kit (Thermo Fisher Scientific) as per manufacturer's instructions, and percentage of EdU$^+$ cells was determined by flow cytometry. The labeling index (LI; percent of total cells positive for EdU) was plotted over time, increasing linearly until a plateau (which corresponds with the GF, or maximum

percentage of cells actively proliferating) was reached. The data can be fitted with 2 regression lines (as described in Nowakowski and colleagues [45]): one line that describes the initial linear increase, and another that describes the horizontal line corresponding to the GF. The time (in the $x$ axis) where the 2 regression lines intersect is equal to Tc − Ts, where Tc is the cell cycle length and Ts is the S phase length. The intersect of the first regression line and the $y$ axis is defined as the initial labeling index (Li0) and is equal to GF × Ts/Tc. Using these equations and the regression lines, Tc and Ts were calculated for WT and $K^{-/-}$ NSCs. Each time point for both genotypes were conducted in duplicates.

## Viral plasmids and viruses

We generated lentiviruses encoding shRNAs targeting endogenous H2-K$^b$ (sh-K), H2-D$^b$ (sh-D), or both (sh-KD) using a lentiviral shRNA expression system (pGreenPuro shRNA, System Biosciences) according to the manufacturer's instructions. The targeted sequences were cloned into the pGreenPuro vector (H2-K$^b$, 5′-GAATGTAACCTTGATTGTTAT-3′; H2-D$^b$, 5′-ACC ACACTCGATGCGGTATTTC-3′; H2-K$^b$ H2-D$^b$, 5′-ACCCTCAGTTCTCTTTAGTCAA-3′), Fgfr1 SH1 (5′-GTGACCGAGGATAACGTAATG-3′), Fgfr1 SH2 (5′-CTGGCTGGAGTCTCC GAATAT-3′), Fgfr1 SH3 (5′-TGAAGACTGCTGGAGTTAATA-3′), Fgfr1 SH4 (5′-CGCTCT ACCTGGAGATCATTA-3′). After knockdown validation, Fgfr1 SH2 was used for further in vitro experiments. Plasmid quality was tested with western blot analysis and Sanger sequencing. For the lentiviral H2-K-1, H2-D-1, and Fgfr1 overexpression constructs, the whole CDS and part of the UTRs were amplified from Hippocampal cDNA and cloned into the pENTR-D--TOPO vector. From there, the CDS was further PCR amplified, and restriction enzyme sites were incorporated into the forward and reverse primers. H2-K1 and H2-D1 CDS were cloned into the Nestin promoter-based lentiviral plasmid [21] using traditional restriction enzyme-based cloning strategy with NheI and BamHI. Fgfr1 was cloned using NotI and SalI. A Nestin-GFP construct based on the same plasmid backbone was used as a control. The final plasmid sequences were verified using Sanger sequencing. LV vectors were generated at the UCSF Viracore, with viral titers between $1.2 \times 10^9$ to $1.5 \times 10^9$ viral particles per mL. Viral solutions were diluted to $1.0 \times 10^8$ viral particles/ml prior to injection. Lentiviruses were stereotaxically injected into the right DG, and a control virus encoding a scrambled shRNA sequence (sh-control, 5′-GGACGAACCTGCTGAGATAT-3′) or Nestin-GFP into the left contralateral DG.

## Bioinformatic analysis of single cell RNA-sequencing data from Shin and colleagues [27]

Pseudotime assignment and read counts (CPM) for all cells were obtained from Shin and colleagues [27]. Read counts were log2 transformed (log2(CPM+1)) and scaled between 0 and 1 by dividing the normalized read count for each gene in each cell by the maximum expression value across all cells for the respective gene. Cells were then ordered according to their pseudotime estimate from early to late. Subsequently, a local polynomial regression line was fit alone the trajectory, and the 95% confidence interval was calculated using the ggplot2 R package. Time points T1 and T2 were determined by the time of increasing *Tbr2* expression (rightmost stationary point of the *Tbr2* regression line calculated in R). G1 and G2 cell cycle score were calculated as the summed expression of G1 and G2 markers, as published in Tirosh and colleagues [29], divided by the median score across cells.

## NSC RNA-sequencing library construction

After RNA isolation, RNA-seq libraries were constructed using the Smart-Seq2 protocol from Trombetta and colleagues [46], with modifications. Briefly, 1 ng high-quality RNA was reverse

transcribed using SuperScript II (Life Technologies, 18064–014) with a poly-dT anchored oligonucleotide primer and a template switching oligonucleotide primer that generated homotypic PCR primer binding sites. The cDNA underwent 10 rounds of PCR amplification using KAPA HiFi Hotstart (Kapa Biosystems, KK2601), followed by Ampure bead (Agencourt) cleanup. The quality of the amplified cDNA was tested using qPCR for GAPDH and nucleic acid quantitation. High-quality amplified cDNA (1 ng) was fragmented with the Tn5 transposase from the Illumina Nextera kit (FC-131-1096) to a median size of approximately 500 bp. The fragmented library was amplified with indexed Nextera adapters (FC-131-1002) using 12 rounds of PCR. Final libraries were purified with Ampure beads and quantified using a qPCR Library Quantification Kit (Kapa Biosystems, KK4824). Libraries were pooled for sequencing on an Illumina HiSeq 2500.

## Bioinformatic analysis of NSC RNA-sequencing data

Quality trimming and adapter removal of raw FASTQ files were carried out with cutadapt. Only reads longer than 20 base pairs were kept for downstream analysis. Subsequently, trimmed reads were mapped to the mouse genome (Grchm38) using HISAT2 [47]. Transcripts given by the GENCODEv25 reference (https://www.gencodegenes.org/mouse/) were quantified with featureCounts [48], considering only uniquely aligned reads. Differential expression tests were calculated using Deseq2 [49], and transcripts with an adjusted $p$-value $< 0.05$ were considered differentially expressed. ConsensusPathDB [50] was used for calculation of pathway enrichments. Raw NSC RNA-seq data have been uploaded to Array Express: E-MTAB-5827.

## Statistical analysis

Statistical tests were performed with GraphPad Prism 6 and 8. All data are represented as mean +/− SEM. Differences between treatment conditions were established using an unpaired Student $t$ test (for 2 conditions). For experiments with $> 2$ groups, a one-way ANOVA with a Dunnett's multiple comparison test for comparison to a single control group or a Tukey's multiple comparison test for individual comparison between all groups was used. A two-way ANOVA with Sidak's multiple comparison test was used for western blot analysis with both H2-K1 and Fgf manipulations. $P < 0.05$ was considered statistically significant. Additional statistic details are indicated in the respective figure legends. The numerical data used in all figures are included in in the Supporting information (S1 Data).

## Supporting information

**S1 Fig. Single-cell transcriptomic characterization of Tbr2 and GFAP expression during neurogenesis.** The neurogenic lineage was divided into an early time point (T1) where NSCs express GFAP, and a later time point (T2) where differentiating intermediate neural progenitors begin to express Tbr2. (A) Waterfall plot illustrates the relative expression of *Tbr2* and *GFAP* along a trajectory of adult neurogenesis, as described in Fig 1A. The cutoff point between time points T1 and T2 was determined by the time of increasing Tbr2 expression (last stationary point of the Tbr2 regression line). Data used to generate this figure can be found in the Supporting information Excel spreadsheet (S1 Data). GFAP, glial fibrillary acidic protein; NSC, neural stem cell; Tbr2, T-box brain protein 2.
(PDF)

**S2 Fig. Characterization and validation of MHC I knockout, knockdown, and overexpression.** Quantitative RT-PCR of H2-K$^b$ (left panel) and H2-D$^b$ (right panel) mRNA from

primary NSPCs isolated from WT, K$^{-/-}$, and D$^{-/-}$ knockout mice (A). $n$ = 3 replicates/group. Quantitative RT-PCR of H2-K$^b$ (left panel) and H2-D$^b$ (right panel) mRNA from primary NSPCs isolated from WT and KD$^{-/-}$ double knockout mice (B). $n$ = 3 replicates/group. Quantitative RT-PCR of H2-K$^b$ (left panel) and H2-D$^b$ (right panel) mRNA from primary WT NSPCs infected with lentiviruses encoding shRNA targeting H2-K$^b$ (sh-K) or H2-D$^b$ (sh-D) or luciferase control (sh-Ctrl) 72 hours after infection (C). $n$ = 4 replicates/group. Quantitative RT-PCR of H2-K$^b$ (left panel) and H2-D$^b$ (right panel) mRNA from primary WT NSPCs infected with lentiviruses encoding shRNA concomitantly targeting both H2-K$^b$ and H2-D$^b$ (sh-KD) or luciferase as a control (sh-Ctrl) (D). $n$ = 4 replicates per group. Quantitative RT-PCR of H2-K$^b$ (left panel) and H2-D$^b$ (right panel) mRNA from primary WT NSPCs infected with lentiviruses overexpressing H2-K$^b$ (OE-K), H2-D$^b$ (OE-D), or GFP as a control (OE-ctrl) under the Nestin promoter (C). $n$ = 4 replicates/group. All data represented as mean ± SEM; one-way ANOVA with Dunnett's post hoc test (A, C, E) and Student $t$ test (B, D); ****$p$ < 0.0001. Data used to generate this figure can be found in the Supporting information Excel spreadsheet (S1 Data). MHC I, major histocompatibility complex class I; NSPC, neural stem and progenitor cell; RT-PCR, reverse transcription polymerase chain reaction; shRNA, short hairpin RNA; WT, wild-type.
(PDF)

**S3 Fig. Characterization of hippocampal composition in MHC I knockout mouse models.** (A) Quantification of Iba1 (a microglial marker) and the percent of microglia coexpressing the activation marker CD68 in the DG of 2–3 months old WT, K$^{-/-}$, and D$^{-/-}$ knockout mice. $n$ = 4 animals/group (3–6 sections/animal). (B, C) Quantification of GFAP-positive astrocytes (B) or overall DG size using the neuronal marker NeuN (C). $n$ = 4–5 animals/group (3–4 sections/animal). All data represented as mean ± SEM. ANOVA with Dunnett's post hoc test. Data used to generate this figure can be found in the Supporting information Excel spreadsheet (S1 Data). CD68, cluster of differentiation 68; DG, dentate gyrus; GFAP, glial fibrillary acidic protein; Iba1, ionized calcium-binding adapter molecule 1; MHC I, major histocompatibility complex class I; NeuN, neuronal nuclei; n.s., not significant; WT, wild-type.
(PDF)

**S4 Fig. Neurogenesis is increased in H2-K$^b$/H2-D$^b$ double knockout mice, and proliferation is increased in primary NSPCs.** (A) Neurogenesis was characterized in 2-month-old WT and H2-K$^b$/H2-D$^b$ double-knockout (KD$^{-/-}$) mice. Quantifications of Nestin+ NSCs, BrdU + short-term proliferating cells, Dcx+ neuroblasts, and adult-born neurons coexpressing BrdU-long and NeuN in the DG. $n$ = 3–5 animals/group (3–6 hippocampal sections/animal). (B) Quantification of Iba+ microglia and the percent of Iba+ microglia expressing CD68 in the DG of adult (2 months) WT or KD$^{-/-}$ mice. $N$ = 4 animals/group (3–6 hippocampal sections/animal). (C) Primary NSPCs isolated from the hippocampi of WT, KD$^{-/-}$ mice were cultured under self-renewal conditions and treated with EdU for 6 hours. Percentage of EdU$^+$ cells are shown (C). $n$ = 3 replicates/group. (D) WT postnatal (left panel) and adult (right panel) NSPCs infected with lentiviruses encoding sHRNA concomitantly targeting H2-K$^b$ and H2-D$^b$ (sh-KD) or targeting luciferase as a control (sh-Ctrl) were cultured under self-renewal conditions and treated with EdU for 6 hours. Percentage of EdU$^+$ cells are shown (D). $n$ = 3 replicates/group. All data represented as mean ± SEM. Student $t$ test; *$p$ < 0.05. Data used to generate this figure can be found in the Supporting information Excel spreadsheet (S1 Data). BrdU, 5-bromo-2′-deoxyuridine; CD68, cluster of differentiation 68; Dcx, Doublecortin; DG, dentate gyrus; EdU, 5-ethynyl-2′-deoxyuridine; Iba1, ionized calcium-binding adapter molecule 1; NeuN, neuronal nuclei; n.s., not significant; NSC, neural stem cell; NSPC, neural stem

and progenitor cell; shRNA, short hairpin RNA; WT, wild-type.
(PDF)

**S5 Fig. No apparent alteration in neuronal differentiation potential in NSPCs lacking H2-K$^b$ or H2-D$^b$.** WT, K$^{-/-}$, and D$^{-/-}$ NSPCs were cultured under growth factor-free conditions and allowed to differentiate for 6 days. Representative field (A) and quantification (B) of Map2$^+$ neurons and GFAP$^+$ astrocytes. Data represented as mean percentages ± SEM; $n = 3$ replicates per group; ANOVA, with Dunnett's post hoc test (B). (C) WT adult NSPCs were infected with shRNA lentiviruses against H2-Kb (sh-K), H2-Db (sh-D), or luciferase (sh-Ctrl). Seventy-two hours after infection, NSPCs were transitioned to growth factor-free culturing conditions and allowed to differentiate for 6 days. Quantification (C) of Map2$^+$ neurons (left panel) and GFAP$^+$ astrocytes (right panel). All data represented as mean percentages ± SEM; $n = 8$ replicates/group. ANOVA with Dunnett's post hoc test. Data used to generate this figure can be found in the Supporting information Excel spreadsheet (S1 Data). GFAP$^+$, glial fibrillary acidic protein-positive; Map2$^+$, microtubule-associated protein 2-positive; n.s., not significant; NSPC, neural stem and progenitor cell; shRNA, short hairpin RNA; WT, wild-type.
(PDF)

**S6 Fig. Loss of H2-K$^b$ in NSPCs alters cell cycle dynamics.** (A, B) A publicly available dataset of single adult hippocampal NSCs [27] was sorted by expression of G1 (x-axis) and G2 (y-axis) cell cycle scores as described in Tirosh and colleagues [29]. Cells positive for H2-K1 expression are indicated in red (A). (B) The percentage of cycling cells was found to be lower in *H2-K1*-positive cells (5.6%, 1 cycling cell, 17 noncycling cells) versus *H2-K1*-negative cells (24.4%, 20 cycling cells, 62 noncycling cells); odds ratio: 0.18. (C) Primary hippocampal WT and K$^{-/-}$ NSPCs cultured under self-renewal conditions were treated with EdU at 2, 4, 6, 8, 10, 12, 14, 16, 18, 20, 22, or 24 hours prior to fixation, and percentage of EdU$^+$ cells was determined by flow cytometry. Percent of cells incorporating EdU reached a plateau at 13.47 hours for WT NSPCs and 13.24 hours for K$^{-/-}$ NSPCs (Tc − Ts) when the cycling population (GF) entered the S phase. Linear regression for increasing phase of WT NSPCs: $y = 0.6133 \times + 7.996$; $R^2 = 0.829$ and K$^{-/-}$ NSPCs: $y = 1.165 \times + 7.406$; $R^2 = 0.889$. Values for cell cycle parameters calculated from cumulative labeling are shown in the right panel. (C). $n = 3$ replicates per group for in vitro experiments. Data represented as mean ± SEM; two-way repeated measures ANOVA with Sidak's post hoc test (K); $^*p < 0.05$, $^{**}p < 0.01$, $^{***}p < 0.001$. Data used to generate this figure can be found in the Supporting information Excel spreadsheet (S1 Data). EdU, 5-ethynyl-2′-deoxyuridine; GF, growth fraction; NSC, neural stem cell; NSPC, neural stem and progenitor cell; Tc, length of cell cycle; Ts, length of S phase; WT, wild-type.
(PDF)

**S7 Fig. Related to Fig 3. Loss of H2-K$^b$ induces gene expression changes in NSPCs.** RNA-seq analysis of primary hippocampal WT, K$^{-/-}$, and D$^{-/-}$ NSPCs cultured under self-renewal conditions. (A) Venn diagram illustrating the number of genes differentially expressed (DESeq2 adj. $p < 0.05$) between WT, K$^{-/-}$, and D$^{-/-}$ NSPCs. (B) Scatterplot of the average gene expression in WT (x-axis) and K$^{-/-}$ (y-axis) NSPCs. Differentially expressed genes (DESeq2 adj. $p < 0.05$, 432 down- and 408 up-regulated) are indicated in red. Genes shown in heatmap (Fig 3B) are colored in blue. Data used to generate this figure can be found in the Supporting information Excel spreadsheet (S1 Data). NSPC, neural stem and progenitor cell; RNA-seq, RNA-sequencing; WT, wild-type.
(PDF)

**S8 Fig. Related to Fig 3. Characterization of growth factor receptor expression in H2-K$^b$-deficient NSPCs.** WT adult NSPCs were infected with either sh-K, sh-D, and sh-control (sh-

Ctrl) or Nestin-driven H2-K$^b$ (OE-K), H2-D$^b$ (OE-D), and GFP control (OE-Ctrl) overexpression lentiviruses. NSPCs were grown under self-renewal conditions, and Fgfr2 (A), Ntrk2 (B), and Egfr (C) expression was measured using quantitative RT-PCR 72 hours after infection. $n = 4$ replicates/group. All data represented as mean ± SEM; one-way ANOVA with Dunnett's multiple comparisons test; $^*p < 0.05$; $^{***}p < 0.001$. Data used to generate this figure can be found in the Supporting information Excel spreadsheet (S1 Data). Egfr, epidermal growth factor receptor; Fgfr2, fibroblast growth factor receptor 2; GFP, green fluorescent protein; n.s., not significant; NSPC, neural stem and progenitor cell; Ntrk2, neurotrophic tyrosine kinase receptor type 2; RT-PCR, reverse transcription polymerase chain reaction; WT, wild-type.
(PDF)

**S9 Fig. Characterization and validation of Fgfr1 knockdown and overexpression lentiviral tools.** (A) WT adult NSPCs were infected with 4 different shRNAs targeting *Fgfr1* (sh-Fgfr1) or sh-control (sh-Ctrl) lentiviruses. NSPCs were grown under self-renewal conditions, and Fgfr1 expression was measured using quantitative RT-PCR 72 hours after infection. sh-Fgfr1 (2) was selected for additional experiments. $n = 4$–5 replicates/group. (B) WT adult NSPCs were infected with a Nestin-driven *Fgfr1* (OE-Fgfr1) or GFP control (OE-Ctrl) overexpression lentiviruses. NSPCs were grown under self-renewal conditions, and Fgfr1 expression was measured using quantitative RT-PCR 72 hours after infection. All data represented as mean ± SEM; one-way ANOVA with Dunnett's multiple comparisons test (A) and Student $t$ test (B); $^{**}p < 0.01$; $^{****}p < 0.0001$. Data used to generate this figure can be found in the Supporting information Excel spreadsheet (S1 Data). Fgfr1, fibroblast growth factor receptor 1; GFP, green fluorescent protein; NSPC, neural stem and progenitor cell; RT-PCR, reverse transcription polymerase chain reaction; shRNA, short hairpin RNA; WT, wild-type.
(PDF)

**S10 Fig. Inhibition of Fgfr signaling restores proliferation in H2-K$^b$-deficient NSPCs.** WT postnatal (D) and adult (E) NSPCs were infected with lentiviruses encoding sh-K and sh-control (sh-Ctrl), cultured under self-renewal conditions and exposed to Fgfr inhibitor PD 161570 (PD, 0.1 uM) or phosphate-buffered saline control for 1 hour and subsequently treated with EdU for 6 hours to test for proliferation (D, E). Data represented as mean percentage of EdU$^+$ cells ± SEM. $n = 3$ replicates/group. Student $t$ test; $^*p < 0.05$ and $^{**}p < 0.01$. Data used to generate this figure can be found in the Supporting information Excel spreadsheet (S1 Data). EdU, 5-ethynyl-2′-deoxyuridine; Fgfr, fibroblast growth factor receptor; n.s., not significant; NSPC, neural stem and progenitor cell; WT, wild-type.
(PDF)

**S1 Data. Numerical data used to generate plots in all figures.** This Excel file contains multiple sheets, each of which contains the data that were used to generate plots in Figs 1–3 and S1–S10 (separated into different sheets for each figure panel).
(XLSX)

**S1 Raw Images. Uncropped western blots data related to Fig 3F.**
(PDF)

**S1 Table. List of reagents and tools used in this study.**
(PDF)

## Acknowledgments

We thank Dr. Shelly Fan for critically reading manuscript. We thank Dr. Carla J. Shatz for generously providing invaluable advice.

## Author Contributions

**Conceptualization:** Karin Lin, Gregor Bieri, Saul A. Villeda.

**Data curation:** Karin Lin, Gregor Bieri, Geraldine Gontier.

**Formal analysis:** Karin Lin, Gregor Bieri, Geraldine Gontier, Sören Müller.

**Funding acquisition:** Saul A. Villeda.

**Investigation:** Karin Lin, Gregor Bieri, Geraldine Gontier, Lucas K. Smith, Cedric E. Snethlage, Charles W. White, III, Sun Y. Maybury-Lewis.

**Project administration:** Saul A. Villeda.

**Supervision:** Saul A. Villeda.

**Validation:** Geraldine Gontier.

**Writing – original draft:** Karin Lin, Saul A. Villeda.

**Writing – review & editing:** Gregor Bieri.

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
