## [Editor Report · Decision Letter 0]

17 Aug 2020

Dear Dr Villeda, 

Thank you for submitting your manuscript entitled "MHC CLASS I H2-KbNEGATIVELY REGULATES NEURAL PROGENITOR CELL PROLIFERATION BY INHIBITING FGFR SIGNALING" for consideration as a Short Report by PLOS Biology.

Your manuscript has now been evaluated by the PLOS Biology editorial staff, as well as by an Academic Editor with relevant expertise, and I am writing to let you know that we would like to send your submission out for external peer review. The Academic Editor thinks, however, that it would be good if your move Fig. S4 to the main figures.

In addition, before we can send your manuscript to reviewers, we need you to complete your submission by providing the metadata that is required for full assessment. To this end, please login to Editorial Manager where you will find the paper in the 'Submissions Needing Revisions' folder on your homepage. Please click 'Revise Submission' from the Action Links and complete all additional questions in the submission questionnaire.

Please re-submit your manuscript within two working days, i.e. by Aug 19 2020 11:59PM.

Kind regards,

Gabriel Gasque, Ph.D.,

Senior Editor

PLOS Biology

---

## [Decision Letter · Decision Letter 1]

17 Sep 2020

Dear Dr Villeda,

Thank you very much for submitting your manuscript "MHC CLASS I H2-KbNEGATIVELY REGULATES NEURAL PROGENITOR CELL PROLIFERATION BY INHIBITING FGFR SIGNALING" for consideration as a Short Report at PLOS Biology. Your manuscript has been evaluated by the PLOS Biology editors, by an Academic Editor with relevant expertise, and by three independent reviewers.

In light of the reviews (below), we will not be able to accept the current version of the manuscript, but we would welcome re-submission of a much-revised version that takes into account the reviewers' comments. We cannot make any decision about publication until we have seen the revised manuscript and your response to the reviewers' comments. Your revised manuscript is also likely to be sent for further evaluation by the reviewers.

We expect to receive your revised manuscript within 3 months. 

**IMPORTANT - SUBMITTING YOUR REVISION**

Your revisions should address the specific points made by each reviewer. As you will see, all three reviewers agree that the study has potential for PLOS Biology. However, they also suggest a number of additional experiments and controls aimed at strengthening your core conclusion that H2-Kb negatively regulates NSPC proliferation and adult hippocampal neurogenesis. Additionally, there were concerns raised that the suggestion of FGFR signaling mechanism would also need to be bolstered with additional experimental support. Having discussed these comments with the Academic Editor, we think they all should be thoroughly addressed. However, we appreciate that to complement the genetic mutant model, and to circumvent potential developmental effects of a constitutive knockout, you have already used an in vivo viral-mediated RNAi approach. Therefore, we do not think this point is critical.

Please submit the following files along with your revised manuscript:

*Re-submission Checklist*

*Published Peer Review*

*PLOS Data Policy*

*Blot and Gel Data Policy*

Sincerely,

Gabriel Gasque, Ph.D.,

Senior Editor,

ggasque@plos.org,

PLOS Biology

REVIEWS:

Reviewer's Responses to Questions

Reviewer #1: In their work Li et al. investigate the MHC molecule H2K1 as a regulator of proliferation during hippocampal neurogenesis. Using a mix of mouse genetics, in vitro analyses, expression analyses and biochemistry they provide evidence that H2K1 acts as a negative regulator of DG neurogenesis, acting not directly on the stem cells, but at later stages. They also claim the this action is mediated by the FGFR pathway. 

The paper is well written. The mouse genetics approach is well designed and the analyses of these animals are convincing. The use of RNAi represents a good support for the mouse mutation analyses. Overall this is an interesting paper that has potential for publication in PLOSBio. However, there are some issues, in particular in the last FGFR-related part of the manuscript, that dampen my enthusiasm and that should be addressed. 

First, as said, the loss of function studies in Fig. 1 are well performed and demonstrate that H2K1 is necessary to dampen DG neurogenesis. It would be interesting, and nicely complementary, to see what happens if the molecule is overexpressed and/or maintained at later stages in stem cells, in a gain of function approach. Is it sufficient to decrease proliferation? Is downregulation of H2K1 necessary? As the authors use the lentiviral approach for the RNAi study, this should not be too difficult. 

Second, the expression, biochemical and pharmacological data in Fig. 3 appears fragmentary and unclear. The expression analysis shows that several GF receptors (FRGG1-3 /EGFR/Ntrk2) are expressed in NPSCs, but only FGFR1 is then presented in detail in the mutant contexts (3b). This data would be far more convincing with a systematic analysis of the expression of all GFRs in the mutants. Moreover, if only FGFR is a candidate, why is there EGF in the GF-mix in Fig. 3C. The pharmacological approach is too limited. Again, a systematic and complete analysis using several pharmacological interventions would be necessary to demonstrate specificity of the FGFR pathway. To my eyes this entire part is rather correlative and indirect.

Reviewer #2: In present study, Lin and co-authors studied the roles of major histocompatibility complex class I (MHC I) molecules, H2-K and H2-D in regulation of neural stem/progenitor cells and adult hippocampal neurogenesis. Specifically, they identified that H2-K, but not H2-D, governed adult neurogenesis by regulating cell cycle dynamics. Using hippocampal NSPCs culture combined with RNA-sequencing, they found that H2-K regulate cell proliferation through inhibiting FGFR-mediated signaling pathway. In general, it is a potential interesting study in the field. But there are still some major concerns that require further attention.

1) The in vivo neurogenesis has been performed on H2-K KO mice. However, the authors have not assessed whether H2-K mutation has developmental consequences on the DG cellular composition, which is highly possible if H2-K is a major actor in neurogenesis. In addition, H2-K deficiency led to increased proliferation of postnatal-derived NSPCs in vitro (Fig. 2B). Therefore, the defective adult hippocampal neurogenesis in H2-K KO mice could be a consequence of developmental defect, but is not specific in adult brain. 

2) Although the bioinformatics analysis showed that the expression of H2-K was expressed in NSPCs and suggested its potential role in regulating adult hippocampal neurogenesis, it is still lacking of the in vivo expression pattern of H2K in adult hippocampal neurogenic niche to argue the specific role of H2K in NSPCs.

3) In adult K-/- mice, more generated new-born neurons (DCX+ cells) are observed, while there is an increased proliferation (Edu+ cells) in cultured K-/- NPSC, which suggested that the increased neurogenesis might be resulted by enhanced proliferation. It is necessary to perform short-term Edu/Brdu or use Ki67/MCM2 combined with cell type specific markers to evaluate proliferative status in vivo. 

4) Both KD-/- and K-/- mice displayed increased neurogenesis in the adult hippocampus (Fig. 1 D and Fig. 1E, DCX and NeuN). However, it seems the number of Nestin+ cells in KD-/- mice is higher than that in K-/- mice (~3000 Nestin+ cells KD-/- mice [Fig.1E], while ~2000 Nestin+ cells K-/- mice [Fig. 1D]). Further comparison the number of Nestin+ cells between K-/- and KD-/- should be done to conclude whether the defective neurogenesis is due to altering Nestin+ cells in these two mutations.

Reviewer #3: In manuscript PBIOLOGY-D-20-02289, Lin et al. address the important issue of whether MHCI molecules regulate neural stem cell proliferation and neurogenesis. The authors first performed transcriptomic analyses of Neural Stem and Progenitors Cells (NSPCs) by analyzing a publicly available dataset of Nestin positive cells and their immediate progeny. Through this analysis, Lin et al found that the class I MHC H2-Kb was downregulated in NSCs that were more neurogenic in their transcriptomic signature, while H2-Db did not change significantly. The authors follow up with several of loss of function experiments with knockout and knockdown of the 2 Class I MHC's (H2Kb, H2Db) both separately and together, that implicate H2Kb in repressing the proliferation of NPSCs. This change in proliferation is described as an increase in DCX + and NeuN/Brdu+ cells in the DG of the adult hippocampus. In vitro experiments using primary NPSCs from both postnatal and adult mice showed that the % of EDU+ cells increases significantly when H2K is knocked out or knocked down with RNAi. Finally, transcriptomic analysis of NSPCs from H2K-/- mice showed a large difference in the transcriptomic signature compared to WT and H2D-/- cells. Specifically, cells that have elevated levels of H2K are less likely to have a cycling (proliferative) transcriptomic signature and FGFR1 mRNA is upregulated in K-/- NSPCs. They also found that an FGFR pharmacological inhibitor reduces the large increase in the % of Edu+ proliferating cells caused by H2K knoc down. The authors conclude that MHC I is required to regulate NSPC proliferation through inhibiting FGFR1 signaling. 

Although these data are novel and address an important gap in our knowledge, there are several concerns that need to be addressed.

(1) The Western blot experiments should have been performed in the presence of phosphatase inhibitors to preserve the phosphorylation status of these proteins after lysis. There is also no quantification displayed, which is critical to assess any significant differences after 

normalization of pERK to ERK and ERK and other proteins to GAPDH or Actin. Finally, the authors should indicate how many repeats of this biochemical experiment were performed; they are typically perfoemed in triplicate for quantification.

(2) It is unclear if the effects of MHCI on proliferation are truly specific and/or limited to H2K? Given that there is residual H2D mRNA expression in the H2D knockout, and the nominally higher levels of proliferative cells in the KD-/-, would it be more accurate to say that there remains a possibility of H2D having a role (although limited) in NSPC proliferation? One experiment to address this could be to express H2K and H2D In knockout K-/- KD-/- primary NSPCs, expecting to see a reduction in proliferation. Can H2D replace H2K to inhibit proliferation in this context? 

(3) Data should be plotted by dot plot rather than bar graphs so that the variability can be more easily assessed.

(4) Figure 1: Panels D and E. The values for WT Dcx+ cells (~5K vs. ~10 K) and NeuN+BrDU+ cells (~600 vs. ~1200) are very different between the 2 experiments, whereas the control numbers of nestin+ cells are similar (~28K in both experiments). This raises the question of whether the WT used for each comparison are the appropriate controls, or whether the WT values are driving the result. The authors should clarify if the WT mice used for the comparisons to the single KO lines were littermates or a separate line and if the latter, they should indicate how many generations apart the lines are and include some data from littermates to increase confidence that the WT values are indeed the appropriate control comparison.

(5) More details about the samples used for the experiments should be included. When sample size is presented as n = 4, for example, the authors should explain whether there are 4 sections or 4 mice used, and whether the mice were from separate litters.

(6) On page 5, the authors state that "Together, these data suggest the influence of H2-Kb molecules in the adult hippocampal neurogenic niche is exerted during later stages of neurogenesis rather than the stem cell pool. " But, doesn't the increase in Edu+ cells in panel F, Figure 1, as well as other data in the paper indicate that H2-Kb regulates early stages of proliferation? 

(7) Figure S4 is interpreted as demonstrating that there is no change in the % of neurons or astros in the cultures generated from WT, K-KO or D-KO mice. Yet, it looks like there might be a decrease in the # of cells from the D-KO, which would be important if true since a main conclusion from this paper is that H2-D does not play a role in adult neurogenesis. Including the actual data plotted (average value +/- sem, and the p-value) and increasing sample size for at least the controls in this particular experiment would determine if this trend is real. While it could be argued that increasing sample size may not be warranted due to power analysis, the actual variability in the experiment indicates that additional samples should be included to make the conclusion that H2-D does not affect proliferation.

(8) The approach used to determine whether cells express H2-K, or not, in Figure 2 seems potentially arbitrary. The authors should include a justification for selecting the threshold of 5 CPM as positive or negative for expression.

(9) On page 6, the authors state that "These bioinformatics data provide evidence that H2-Kb regulates NSPC proliferation in a cell-intrinsic manner, potentially by impacting the cell cycle." However, it is unclear why they conclude that this is cell-intrinsic. 

(10) Figure 3, panel B. The graph appears to show a significant decrease in Fgfr1 expression in H2-D KO cells that is not indicated by an asterisk. This decrease would be important to note, if true, given the clear conclusion by the authors that H2-D does not regulate proliferation.

(11) Figure 3, panels D and E. Although the FGFR inhibitor largely rescues the sh-K-induced increase in proliferation in adult neurons, it is less effective in postnatal neurons. The authors should discuss potential reasons why this is the case.

(12) The authors interpret the result in Figure 3 as showing that "H2-Kb molecules negatively regulate NSPC proliferation by inhibiting FGFR-mediated signaling." Yet, the authors do not show this directly. To do so, they would need to increase H2Kb levels, show a decrease in FGFR signaling and in proliferation, and then rescue both by normalizing FGFR signaling. 

Minor:

(1) Fig1. Legend- refers to S2, should be reference to S1.

(2) Fig2. Panel F X-axis should be labeled. J) Provide the number of cells in each group somewhere (H2-K1 positive, H2-K1 negative)

(3) Fig3. B) make clear in axis label that this is RNA 

(4) Typo "ribosomal Proteign" in figure legend

Methods 

(5) AG 99 is referenced in inhibitor experiments but it is unclear in what experiments it is used. 

(6) Methods include references to western blot quantification, but none is presented in the paper

(7) Details are missing about how long after lentiviral injection for knockdown experiments BRDu was injected and animals perfused.

(8) C57 Bl6N(Tac) and Bl6J are both listed in methods. Which strain was a control for which experiments?

---

## [Decision Letter · Decision Letter 2]

20 May 2021

Dear Dr Villeda,

Thank you for submitting your revised Short Report entitled "MHC CLASS I H2-KbNEGATIVELY REGULATES NEURAL PROGENITOR CELL PROLIFERATION BY INHIBITING FGFR SIGNALING" for publication in PLOS Biology. I have now obtained advice from the original reviewers and have discussed their comments with the Academic Editor. 

Based on the reviews, we will probably accept this manuscript for publication, provided you satisfactorily address the data and other policy-related requests listed below my signature. 

We expect to receive your revised manuscript within two weeks. 

*Published Peer Review History*

*Early Version*

Sincerely,

Gabriel Gasque, Ph.D.,

Senior Editor,

ggasque@plos.org,

PLOS Biology

TITLE:

To avoid confusion at copy-edit, please remove all caps from your title, reserving the capital letters for the words that need them: MHC class I H2-Kb, FGFR.

COMPETING INTEREST:

Please include the following statement when declaring a competing interest: "LS is an associate editor for PLOS Biology. He was blinded to this manuscript and all related information in the journal's submission system and not involved at all in editorial discussions or the peer review process."

ETHICS STATEMENT:

-- Please include the ID number of your experimental protocol approved by the University of California San Francisco IACUC.

-- Please include the specific national or international regulations/guidelines to which your animal care and use protocol adhered. Please note that institutional or accreditation organization guidelines (such as AAALAC) do not meet this requirement.

DATA POLICY:

Thank you for providing the raw NSC RNA-sequencing data via ArrayExpress. 

In addition, we need you to provide all individual quantitative observations that underlie the data summarized in the figures and results of your paper. For an example see here: http://www.plosbiology.org/article/info%3Adoi%2F10.1371%2Fjournal.pbio.1001908#s5

These data can be made available in one of the following forms:

Regardless of the method selected, please ensure that you provide the individual numerical values that underlie the summary data displayed in the following figure panels: Figures 1ABEGHI, 2C-E, 3ABDEFGH, S1A, S2A-E, S3AB, S4A-D, S5BC, S6A-C, S7B, S8A-C, S8AC, and S10AB.

Please also ensure that each figure legend in your manuscript includes information on where the underlying data can be found and that your supplemental data file/s has/have a legend.

We require the original, uncropped and minimally adjusted images supporting all blot and gel results reported in an article's figures or Supporting Information files. We will require these files before a manuscript can be accepted so please prepare and upload them now. Please carefully read our guidelines for how to prepare and upload this data: https://journals.plos.org/plosbiology/s/figures#loc-blot-and-gel-reporting-requirements 

DATA NOT SHOWN?

Please note that per journal policy, we do not allow the mention of "data not shown", "personal communication", "manuscript in preparation" or other references to data that is not publicly available or contained within this manuscript. Please either remove mention of these data or provide figures presenting the results and the data underlying the figure(s).

Reviewer remarks:

Reviewer #1: The authors followed my recommendations and provided convincing gain-of-function studies, that nicely complement the loss-of-function approach. Also, the strengthened the link between H2-K1 and Fgfr1 expression with new and convincing data. I have no further critique.

Reviewer #2: The authors have addressed all my concerns.

Reviewer #3: In their resubmission of PBIOLOGY-D-20-02289R1, the authors present a thorough response to all of the critiques and the manuscript is much improved. The only experiment they did not perform was to determine if FGFR overexpression rescues the decrease in proliferation caused by H2Kb overexpression. They have the constructs to perform this experiment, so it's unclear why they couldn't complete it. That being said, they do show strong evidence that H2-K1 and Fgfr1 signaling is linked and regulates NSPC proliferation. That is a novel and important finding and I am convinced that the manuscript is now strong enough for publication.

---

## [Editor Report · Decision Letter 3]

4 Jun 2021

Dear Dr Villeda,

On behalf of my colleagues and the Academic Editor, Angélique Bordey, I am pleased to say that we can in principle offer to publish your Short Report "MHC class I H2-Kb negatively regulates neural progenitor cell proliferation by inhibiting FGFR signaling" in PLOS Biology, provided you address any remaining formatting and reporting issues:

>> Please ensure that each figure legend in your manuscript includes information on where the underlying data can be found (S1 Data) 

>> Please ensure that your supplemental data file (S1 Data) has a legend.

>> Please ensure that your Data Statement in the submission system accurately describes where your data can be found, by including "All quantitative data can be found in the supporting material"

In addition, other formatting and reporting requests will be detailed in an email that will follow this letter and that you will usually receive within 2-3 business days, during which time no action is required from you. Please note that we will not be able to formally accept your manuscript and schedule it for publication until you have made the required changes.

PRESS

Thank you again for supporting Open Access publishing. We look forward to publishing your paper in PLOS Biology. 

Sincerely, 

Gabriel Gasque, Ph.D. 

Senior Editor 

PLOS Biology